# A Delphi Survey Study to Formulate Statements on the Treatability of Inherited Metabolic Disorders to Decide on Eligibility for Newborn Screening

**DOI:** 10.3390/ijns9040056

**Published:** 2023-10-11

**Authors:** Abigail Veldman, M. B. Gea Kiewiet, Dineke Westra, Annet M. Bosch, Marion M. G. Brands, René I. F. M. de Coo, Terry G. J. Derks, Sabine A. Fuchs, Johanna. M. P. van den Hout, Hidde H. Huidekoper, Leo A. J. Kluijtmans, Klaas Koop, Charlotte M. A. Lubout, Margaretha F. Mulder, Bianca Panis, M. Estela Rubio-Gozalbo, Monique G. de Sain-van der Velden, Jaqueline Schaefers, Andrea B. Schreuder, Gepke Visser, Ron A. Wevers, Frits A. Wijburg, M. Rebecca Heiner-Fokkema, Francjan J. van Spronsen

**Affiliations:** 1Division of Metabolic Diseases, Beatrix Children’s Hospital, University of Groningen, University Medical Center Groningen, 9718 GZ Groningen, The Netherlands; 2Department of Genetics, University of Groningen, University Medical Center Groningen, 9718 GZ Groningen, The Netherlands; 3Department of Human Genetics, Radboud University Medical Center, 6525 GA Nijmegen, The Netherlands; 4Department of Pediatrics, Division of Metabolic Disorders, Emma Children’s Hospital, Amsterdam University Medical Centre, 1105 AZ Amsterdam, The Netherlands; 5Department of Toxicogenomics, Unit Clinical Genomics, MHeNs School for Mental Health and Neuroscience, Maastricht University, 6229 ER Maastricht, The Netherlands; 6Department of Metabolic Diseases, University Medical Center Utrecht, Wilhelmina Children’s Hospital, 3584 EA Utrecht, The Netherlands; 7Department of Pediatrics, Center for Lysosomal and Metabolic Diseases, Erasmus University Medical Center, 3015 GD Rotterdam, The Netherlands; 8Department of Human Genetics, Translational Metabolic Laboratory, Radboud University Medical Center, 6525 GA Nijmegen, The Netherlandsron.wevers@radboudumc.nl (R.A.W.); 9Department of Pediatrics, Maastricht University Medical Center, 6229 HX Maastricht, The Netherlands; 10Department of Pediatrics and Clinical Genetics, Maastricht University Medical Center, 6229 HX Maastricht, The Netherlands; 11Section Metabolic Diagnostics, Department of Genetics, University Medical Center Utrecht, 3584 CX Utrecht, The Netherlands; 12Department of Laboratory Medicine, Laboratory of Metabolic Diseases, University of Groningen, University Medical Center Groningen, 9718 GZ Groningen, The Netherlands

**Keywords:** newborn screening, treatability, treatable, Wilson and Junger criteria, inborn errors of metabolism, inherited metabolic disorders, next-generation sequencing

## Abstract

The Wilson and Jungner (W&J) and Andermann criteria are meant to help select diseases eligible for population-based screening. With the introduction of next-generation sequencing (NGS) methods for newborn screening (NBS), more inherited metabolic diseases (IMDs) can technically be included, and a revision of the criteria was attempted. This study aimed to formulate statements and investigate whether those statements could elaborate on the criterion of *treatability* for IMDs to decide on eligibility for NBS. An online Delphi study was started among a panel of Dutch IMD experts (EPs). EPs evaluated, amended, and approved statements on *treatability* that were subsequently applied to 10 IMDs. After two rounds of Delphi, consensus was reached on 10 statements. Application of these statements selected 5 out of 10 IMDs proposed for this study as eligible for NBS, including 3 IMDs in the current Dutch NBS. The statement: ‘The expected benefit/burden ratio of early treatment is positive and results in a significant health outcome’ contributed most to decision-making. Our Delphi study resulted in 10 statements that can help to decide on eligibility for inclusion in NBS based on *treatability*, also showing that other criteria could be handled in a comparable way. Validation of the statements is required before these can be applied as guidance to authorities.

## 1. Introduction

Newborn screening (NBS) aims to identify disorders early to prevent or significantly reduce morbidity and mortality. To guide the selection of disorders that could qualify for population-based screening, 10 criteria were established by Wilson and Junger (W&J) in 1968 [1]. These W&J criteria and their revised version by the World Health Organization (WHO) in 2008 and 2011 [2,3] have been the gold standard for the principles for public health screening policies ever since and are used to decide on the inclusion of disorders in NBS [1,2,3,4], despite the fact that they were not originally meant for NBS.

Worldwide, the procedure to include disorders in NBS programs and the number of diseases included in NBS programs varies [5,6]. The United States of America and Denmark use the quantitative scoring matrix of the American College of Medical Genetics (ACMG) to decide on the inclusion of disorders for NBS [6]. But even this Recommended Universal Screening Panel (RUSP) [7,8] has not prevented these countries from selecting different disorders [9]. Australia uses its own National Policy Framework Newborn Blood Spot Screening based on the criteria of W&J, which are “amended to suit NBS and the local context in which programs operate” [10,11]. This National Policy framework references the use of RUSP and the governing documents of the United Kingdom and New Zealand. Still, there are national and even regional differences in the selection of disorders screened for [12]. For example, the state of Victoria decided not to screen for galactosemias, while other states do [12]. These differences can partly be explained by the fact that in practice, the choices regarding disorders included in screening programs often depend on financial support, technical and medical knowledge, and/or the personal interest of healthcare professionals, scientists, and other persons involved in policy-making [5,13]. Despite the guidance of the W&J criteria, there is room for different interpretations regarding the choice to screen for disorders. Unfortunately, it is often not clear on what basis a disease has or has not been selected.

The rationale behind selecting disorders becomes increasingly relevant since we are at the beginning of a new era: a genetic-based NBS [9,14,15,16,17,18,19]. Next-generation sequencing (NGS) techniques allow us to include a much higher number of monogenetic disorders compared to any existing NBS program, albeit with unknown sensitivity. To be able to be seen as a suitable test for NBS in general, these techniques need to pass technical, ethical, and financial hurdles first. If these methods prove suitable, we will need to discuss the other W&J for each disorder. One of the most important W&J criteria in this respect is probably: ‘There should be an accepted treatment for patients with recognized disease’. Within the context of genetic screening possibilities, Andermann et al. [2,3] developed an approach to guide genetic screening policy-making. Twenty criteria were designed, including criterion 17 on intervention: “There should be an accepted intervention (ex. prevention, treatment, family planning) that forms part of a coherent management system”. The aspect of an accepted treatment or *treatability* consists of three main requisites: (1) the presence of treatment for this disorder, (2) the approval of this treatment by the FDA/EMA, and (3) financial coverage or reimbursement by standard healthcare. Without *treatability*, no disorder will be approved to be selected for NBS [2,3].

Van Karnebeek et al. reported on the amenability of inherited metabolic disorders (IMDs) in patients presenting with epilepsy or mental retardation [20,21]. However, it may be questioned whether these IMDs meet the criteria on *treatability* in the context of NBS. The literature lacks a clear definition of *treatability* in the context of NBS. Also, recent innovations, like NGS techniques and improved treatment possibilities for many disorders, generate new ideas about what is a treatable disease. To make an attempt to ease the selection process for diseases potentially eligible for NBS, we aimed to create a list of statements in the context of NBS that elaborates on *treatability* using a Delphi approach [22]. Since the current NBS mainly consists of IMDs, we took this group of disorders as an example.

## 2. Materials and Methods

The methodology of our project was based on the Recommendations for the Conducting and REporting of DElphi Studies (CREDES) [22] stated by EQUATOR network.org (accessed on 3 April 2022), and the further available literature reporting on Delphi Studies [23,24,25,26]. The rationale behind the Delphi technique builds on the assumption that the opinion of a group is more valid than an individual opinion [27,28]. This is used to form a consensus or to explore a field beyond the borders of the current knowledge and conceptual world [22], for example, the term “*treatability*”. Different from the Nominal Group Technique or a consensus conference [27], the Delphi technique is an anonymous process in which every opinion will be heard, preventing the dominance of more influential experts in the discussion. Moreover, it provides the possibility to think about answers. Differently from an interview or a regular survey, the constructive nature of the Delphi technique allows for the generation of new ideas, the possibility to respond to other experts and the chance to re-state one’s opinion [22,29], which seems perfect for a medical ethical discussion on the criterion of *treatability* in the context of NBS. 

### 2.1. The General Design of the Delphi Study

Our Delphi study was designed and conducted in three anonymous online survey rounds (Part 1.1, Part 1.2, and Part 2). The aim of Part 1 was to develop statements and reach consensus on these statements that together could define *treatability* in NBS. The aim of Part 2 was to test these statements on 10 IMDs, to gain insight into which statement contributed the most, and investigate whether it is possible to put weight on each statement. Figure 1 illustrates a flowchart of the design of the study.

#### 2.1.1. Part 1

In Part 1.1, the research team (RT: AV, MRH, FJvS), supported by two medical ethicists (EM, WD), formulated 28 statements subcategorized into seven items (Appendix A) based on a literature review on *treatability.* Statements were also inspired by the NEXUS study [17]. The professionals invited to participate in this study were colleagues with known experience in IMDs from the Dutch Advisory Committee Neonatal Screening for IMDs and the project group members involved in the study on NGS as a first-tier approach in NBS in the Netherlands (NGSf4NBS) [18]. Together, they formed the Expert Panel (EP). Invitations were sent by email. The EP consisted of pediatricians for IMDs (N = 17), clinical laboratory geneticists/chemists (N = 8), clinical laboratory geneticists/chemists (N = 1), and pediatric neurologists with experience with IMDs (N = 2). This total of 28 EP members (EPs) were asked to rate the quality of the *treatability* statements on a 1–10 Likert scale (1 = completely disagree, 10 = strongly agree) and to evaluate whether this statement should be added to the final decision matrix, or not. It was encouraged to substantiate choices by commenting on the statements or to add new relevant statements in special open-field comment boxes. In Part 1.2, all results were anonymized and shared via email. Some of the statements could be improved according to the comments made by the EPs in Part 1.1. Again, the EPs were asked to rate every statement on a 1–10 Likert scale and to comment on and improve the statements. EPs were given a response time of one week each for Part 1.1 and Part 1.2. Before the start of the study, the RT decided that statements met consensus if they had a mean of at least 7.0 and a median of at least 7.0, and a mode of at least 7.0. Statements that met consensus proceeded to Part 2. 

#### 2.1.2. Part 2

In Part 2, EPs were asked to evaluate 10 IMDs on their eligibility for NBS using the statements that met the consensus criterion in Part 1. The selection of these 10 IMDs as examples were based on different IMDs selected by the North Carolina Newborn Exome Sequencing for Universal Screening (NEXUS), The BabySeq Project, RUSP, and our selection of IMDs made in the NGSf4NBS project [18]. We chose seven IMDs that were not selected in each of the studies and for which *treatability* may be debatable. Three IMDs already included in the Dutch NBS served as controls. The 10 IMDs selected for this study were: pyridoxine-dependent epilepsy (PDE; OMIM 266100), classic galactosemia (CG; OMIM 606999), carnitine palmitoyltransferase 2 deficiency (CPT2; OMIM 255110, 600649, 608836), glycogen storage disease type 2 (Pompe disease; GSD2; OMIM 232300), autosomal recessive guanosine triphosphate cyclohydrolase 1 deficiency (GCH1; OMIM 233910), ornithine transcarbamylase deficiency (OTC; OMIM 311250), Wilson’s Disease (WD; OMIM 277900), methylmalonic aciduria due to methyl malonyl-CoA mutase deficiency (MCM; OMIM 251000), tyrosine hydroxylase deficiency (TH; OMIM 605407), and phenylketonuria (PKU; OMIM 261600). For each IMD, three questions were asked: First, to score the IMD on each consented statement on a 0–5 Likert scale (0 not applicable/true, and 5 most applicable/true). Next, to score on a 0–5 Likert scale if the IMD should be included in NBS (0 do not agree, 5 completely agree). An open-field box was added to give EPs the option to elaborate on their views. Finally, the EPs were asked to discriminate between the importance of statements for the final decision matrix. This was achieved via a multiple-choice question to rank which statement(s) contributed most to the decision to include, or not include, this IMD in the NBS. The options were: ‘Did not contribute to my decision at all’ (0 points), ‘Small contribution but should be in the matrix’ (1 point), ‘moderate contribution’ (2 points), and ‘Large contribution (3 points).’ In case an IMD clinically presents with multiple phenotypes, they were asked to reason from the most severe phenotype. For Part 2, the EPs had two weeks to fill in the survey. 

### 2.2. Informational Input and Piloting of Materials

Qualtrics XM, a web-based survey tool, was used (Qualtrics, Seattle 2002). Qualtrics XM (version March 2022) allows for the design, distribution, and processing of online surveys in multiple formats with a professional layout. For every round, a separate survey was created. Therefore, in total, three surveys were created. All responses were anonymous and EPs received an invitation by email with an anonymous link. Every methodological step was thoroughly evaluated by the RT and piloted in advance to test the quality, accessibility, and user-friendliness. 

### 2.3. Strategy for Interpretation and Processing of Results

Between each round, a report of all results was constructed by one of the members of the research team (AV). This report consisted of statistical data, including the mean, median, and mode of all statements, and qualitative comments of the EP per statement and item. All results, including partial responses, were discussed within the RT and the consensus, and decision rules were applied (Table 1). After this, the survey of the new round was created, evaluated, and piloted again within the RT. After approval by all three members of the RT, the Qualtrics XM link for the new round was distributed, together with the result report available for inspection by the EPs via email. 

Part 2 required a strategy for the interpretation of the results and translation into a final decision matrix. First, the EPs looked at the contribution of the statements on a 3-point Likert scale: no contribution (0 points), small contribution (1 point), moderate contribution (2 points), and large contribution (3 points). The statements that contributed most to all IMDs were regarded as important in decision-making and recieved a higher weight in the final list. Second, we analyzed if a disorder should be included in the NBS or not according to the vast majority (>75%) of EPs. Last, we assessed if there was a correlation between an IMD with a high score on the *treatability* statements and an agreement on eligibility for NBS for that IMD.

## 3. Results

### 3.1. Results Part 1.1 and Part 1.2

#### 3.1.1. Panel Participation

Twenty-eight out of forty-one (68.3%) invited Eps filled in the Delphi survey and formed the EP in Table 2. Since the survey was anonymous, the EPs were asked to confirm their participation via email, so that we could send targeted emails for the second and third rounds. Five of them failed to do so. Therefore, only 23 EPs (56%) were invited to participate in Part 1.2 and 2. Of them, 21 EPs (91%) participated in Part 1.2. 

#### 3.1.2. Statements and Item Throughput

Part 1.1 started with seven main items concerning the treatability of disorders. The seven main items included 26 statements on: (Q1) financial reimbursement by insurance/ EMA approval; (Q2) the accepted level of burden or risk for the newborn/child related to the available treatment; (Q3) the risk of disease-related mortality despite treatment; (Q4) the risk of disease-related morbidity despite treatment; (Q5) the outcomes should at least be prevented, reversed, or improved by early detection with consequent treatment through NBS; (Q6) the minimum amount of publications in which this outcome should have been demonstrated; (Q7) the minimum number of patients (both mild and severe variants) in which (any) effect of treatment has been demonstrated, (Appendix A). After ranking the statements in the first round, a total of N= 99 comments was made on the items and statements. At the same time, N = 23 suggestions on the “minimum number of publications” in item Q6.5 or “minimum number of patients” in items Q7.5 were made (Appendix A). After Part 1.1, 16 statements would have reached preliminary consensus based on our criteria. However, for many of them, major adaptations and specifications were requested as side comments. After processing and interpreting the results of Part 1.1, we followed the decision rules of Table 1. Based on that, 9 of 26 statements were removed (two of them were combined with another statement), 5 new statements were added and 7 statements were adapted or specified, while 9 statements were left unchanged, resulting in a total of 22 statements at the end of Part 1.1 (Figure 2). The improved survey was the starting point of Part 1.2.

Part 1.2 started with six questions/items, which were divided into a total of 22 statements (Appendix B). Considerably fewer comments (N = 37) were recorded after this round. Ten statements (45%) reached consensus and proceeded to Part 2. Minor grammatical specifications and rephrasing were needed on the 10 consented statements after Part 1.2. All items and statements of the first two rounds and their alterations can be found in Appendix A and Appendix B. The final list of statements is depicted in Table 2.

### 3.2. Results Part 2

#### 3.2.1. Panel Participation

Twenty-two of twenty-three EPs (95.7%) filled in the survey of Part 2. However, some respondents failed to complete their responses for all 10 proposed IMD, especially due to a self-noted feeling of inadequate knowledge of specific IMDs and due to unknown reasons. The mean number of responses per statement was N = 16.3 (range 11–22) of the total of 23 responses. 

#### 3.2.2. Contribution of Statements

EPs were asked to rate per IMD which statement contributed most to their decision to include a disorder in the NBS or not based only on its *treatability*. The mean contribution score of every statement ranged from 1.8 to 2.6 points (minimum 0.0 and maximum 3.0). Statements 2 (2.6 points), 5 (2.4 points), and 7 (2.4 points) contributed more than the other statements. Statement 10 (1.8 points) contributed the least to the decision-making according to the EP. 

#### 3.2.3. Inclusion in NBS

Table 3 presents an overview of the selected IMDs used to test our 10 statements and the decision to consider these IMDs eligible or not for NBS based on the criterion of *treatability*. This Table also shows whether the IMDs were included in the panels of RUSP [8], and the studies of NEXUS [17], and The BabySeq Project [16,30,31]. According to >75% of the 22 responding EPs, 5 out of 10 IMDs would be added to the NBS, including the 3 disorders that are in the current Dutch NBS (PKU, MCM, and CG). Comments made regarding decision-making in Part 2 are depicted in Appendix C.

#### 3.2.4. The Scoring of Statements per IMD

To assess whether the selection of IMDs for NBS, as described in 6.2.2, correlates with a high score on the *treatability* statements, all mean scores on the 1–5 Likert scale were calculated in Table 4. The first observation was that both PKU and CG, which are already included in the Dutch NBS, scored the highest on almost every statement, resulting in mean scores above 4.0.

For the other IMDs, there was no direct relationship between scoring high on the *treatability* statements and their eligibility for NBS. For example, GSD2 and OTC both scored higher than PDE, MCM, and TH, but were not selected by the EP as eligible for NBS. Additionally, the opinion about the level of *treatability* of some IMDs was strongly divided among respondents. Appendix A presents the variation in the responses of the EPs.

#### 3.2.5. Comments and Evaluation of the EP on the Decision Matrix

The EP was asked an optional open question to give their opinion on whether the statements are a useful tool to quantify *treatability* for individual IMDs in the context of NBS. Twelve of twenty-two EPs (54.5%) responded. Responses varied and could be classified into five main opinions:

Opinion 1:It can be a useful tool if all data are available and experts are involved in decision-making.Opinion 2:It can be a useful tool as a starting point for a discussion for inclusion in NBS.Opinion 3:Concerns about how to proceed with this tool if disorders are not eligible for NBS.Opinion 4:Concerns about how to proceed with this tool for disorders with a broad phenotypic variability.Opinion 5:The decision matrix is not useful (yet) to quantify *treatability* for IMDs in NBS.

A complete overview of all responses can be found in Appendix D.

## 4. Discussion

To the best of our knowledge, this is the first study that designed and performed an online Delphi study with pediatricians and experts on IMDs to formulate statements that help to elaborate on the definition of *treatability* of IMDs in the context of NBS. For this, we introduced a score for *treatability* to assess their eligibility for NBS. In this pioneering study, the statement: “The expected benefit/burden ratio of early treatment is positive and results in a significant health benefit” was found to contribute most clearly to decision-making. The final 10 statements on *treatability* show that a Delphi study with clear consensus and decision rules seems to be a suitable method to create a scoring system on *treatability*. This study also provides more insight into the aspects that IMD experts find important when deciding on the *treatability* of IMDs in the context of NBS, acknowledging the need for validation by international colleagues and patient/parent representatives.

Several methodological aspects should be taken into account before interpreting our results. First, the Delphi technique is one of the most valuable methods to reach consensus on research questions involving a medical ethical discussion [32] and is particularly interesting for topics without clear (data on) consensus [28,32]. This technique is based on expert opinions, which is considered to have the lowest level of evidence in evidence-based medicine according to systems for guideline development [33,34,35,36]. The validity and quality of a Delphi study, however, mainly depend on the design, conduction, analysis, and modification of results between rounds, and the reporting of the Delphi by the investigator, rather than the technique itself [22,37]. By following CREDES [21], and ensuring complete and clear reporting on the entire study, we believe that this study is a first and important step towards a definition of the criterion of *treatability* for NBS. A risk of cognitive biases may occur if the Delphi study is not designed properly due to the formulation and the way in which the survey items and statements are presented [26,38]. We tried to follow the recommendations of Markmann et al., 2021 [38], to minimize the amount of abstract language use and information in surveys, by discussing both the content and the formulation of statement proposals with ethicists (EM, WD). We also tried to prevent framing and anchoring. Framing occurs when the EP is not sufficiently heterogeneous and group-thinking occurs when the EPs have the same background [39,40]. This may lead to a polarized judgment of the statements that is not representative. Framing was at least partly avoided by performing this study anonymously, while the group itself consisted of a wide variety of specialists involved in various aspects of the diagnosis and treatment of IMDs. In a study requiring such specific knowledge of IMDs, only possessed by a few specialists, more extensive heterogeneity is very hard to achieve. Anchoring may occur when the statements are presented in a certain manner or order. We could not easily randomize statements as they, at least in part, refer to each other. Some anchoring could thus not be prevented. Of course, the involvement of IMD professionals can also be considered a strength of this study. The EPs are highly experienced in delivering the best care for IMD patients and are up-to-date on the latest treatment options, which makes them adequate candidates to elaborate on the concept of *treatability*. Therefore, we think that our data are valid. At the same time, validation and weighting of the statements in other groups of professionals, e.g., international colleagues with experience in IMDs and medical ethicists, and patient representative organizations, is needed.

Our statements were primarily based on the W&J criterion 2 and part of the Andermann criterion 17 on *treatability*. No other criteria were considered. On the one hand, we think that each of the criteria deserve to be evaluated in a transparent process. On the other hand, our study shows that it is hard to evaluate each criterion on its own, as *treatability* is highly interrelated to other criteria, such as costs and phenotypic variability (e.g., age of onset, severity). The mean scores also do not take into account the weight of the contribution of the statements. Most information is contained in the details, including the outlier remarks for each statement. These issues could, at least in part, explain the differences seen between the lower-ranking *treatability* scores of PDE, MCM, and TH and the overall conclusion that these disorders are still eligible for NBS, or the opposite, i.e., a high score but not considered eligible for NBS, as in GSD2, OTC, and WD. Likewise, the value of a single statement in the criterion of *treatability* can be low, e.g., statement 6 in GSD2, whereas the statements as a whole provide a more complete impression of the level of *treatability*. The transparency of the process and a highly experienced EP team per IMD are, therefore, essential.

IMDs are (extremely) rare, resulting in a lack of data on their natural history and phenotypic variability, and not all EPs could obtain a complete overview of all IMDs. They may very well impact the EPs’ ideas regarding the aspect of *treatability,* especially if these aspects are not yet addressed individually. The lack of knowledge of natural history and phenotypic variability could also explain the low response rate for some IMDs in Part 2. Between three and eight EPs (13.6–36.4%) failed to respond, stressing the fact that a panel of acknowledged disease experts for each disease is necessary to assess their eligibility for NBS. From this response, it can be concluded that the EPs felt they needed a substantial level of knowledge of an IMD to decide on its *treatability* or eligibility for NBS, knowledge that is limited in the case of rare disorders. Experts on specific IMDs and endocrinological diseases and medical ethicists are needed to ultimately decide on the *treatability* and eligibility of disorders for NBS. Such studies are also needed with parents (to be). This may seem in contrast to the high level of expertise EPs considered necessary to evaluate the statements, but the studies of Armstrong et al., 2022, from the Babyseq project, and the study of Blom et al., 2021, provided at least some evidence that parents do understand the concept of the choice of whether to screen for (un)treatable disorders [41,42,43].

The necessity to obtain a group of disease-specific experts is exemplified by the outcomes of several IMDs. In MCM, scores were relatively low on statements 9 and 10. The discrepancy between agreement on eligibility for NBS and the *treatability* score of MCM might be the result of the differences in severity in the presentation of this IMD; EPs agreed that severe MCM is not fit for inclusion in NBS, while milder MCM variants were found to be eligible, showing that the statements are difficult to assess for IMD, with large phenotypic variances. In PDE, scores are relatively low on statement 4 and statement 6. However, this seems to be due to the opinion of one EP member, who was a clear outlier in the score assessments. This EP member opposed a different approach to this IMD by reasoning that PDE should be included in the guidelines for the treatment of epilepsy rather than including PDE in the NBS. This can also be learned from CPT2. In CPT2, for example, the EP considered more knowledge necessary to be able to judge the severity of disease presentation in order to reduce the potential risk of overtreatment in patients with less severe disease. Based on the comments of the EP, some statements deserve minor improvements, including the addition of a more detailed description of ‘significant health benefits’ (statement 2), the addition of a ‘not-applicable’ option to the statements (e.g., statement 3 is not applicable for PKU), and the addition of consensus on the positive outcome of early detection by an (international) expert meeting (statement 9).

In conclusion, we built a scoring system based on the statements. This Delphi study allowed us to gain insight into *treatability* in the context of NBS. Our study shows that, with solid statements, it is possible to further elaborate on one specific criterion, e.g., *treatability*, to determine the eligibility of disorders for NBS. Our study shows that it is very valuable to have such a discussion on *treatability* in the most transparent way, but also shows that this cannot be achieved without also addressing the other criteria in similar processes. Most criteria are interrelated; therefore, we consider this study as a starting point to help select disorders for NBS in an era in which NGS techniques increase the number of IMDs and other genetic diseases that are technically eligible for NBS. At the same time, we are on the brink of an era in which more and more (genetic) disorders will become increasingly treatable. Since this is the first study, to our knowledge, ever investigating a more transparent process for eligibility for NBS, this approach needs to be fine-tuned. We envision a funnel procedure to evaluate the eligibility of a disease by passing each of the W&J criteria and/or Andermann one by one, following the same procedure for every criterion and every disease.

## Figures and Tables

**Figure 1 IJNS-09-00056-f001:**
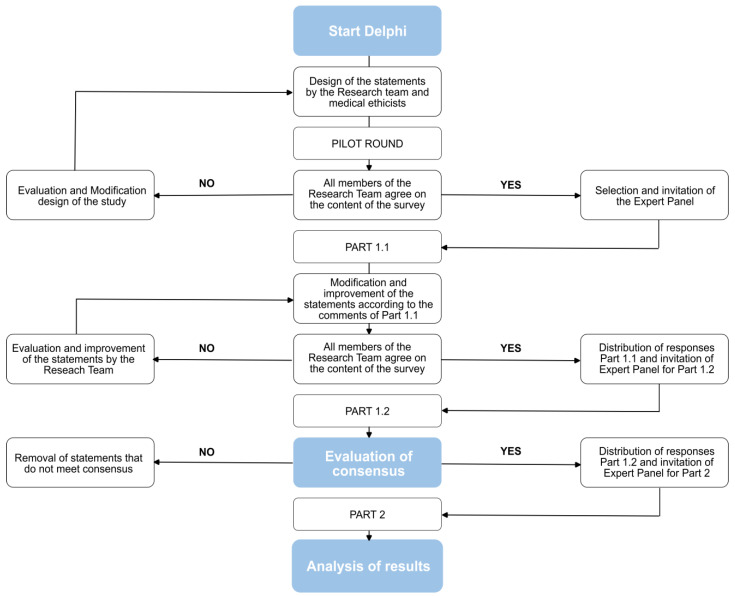
Flowchart of the design of the Treatability Delphi study.

**Figure 2 IJNS-09-00056-f002:**
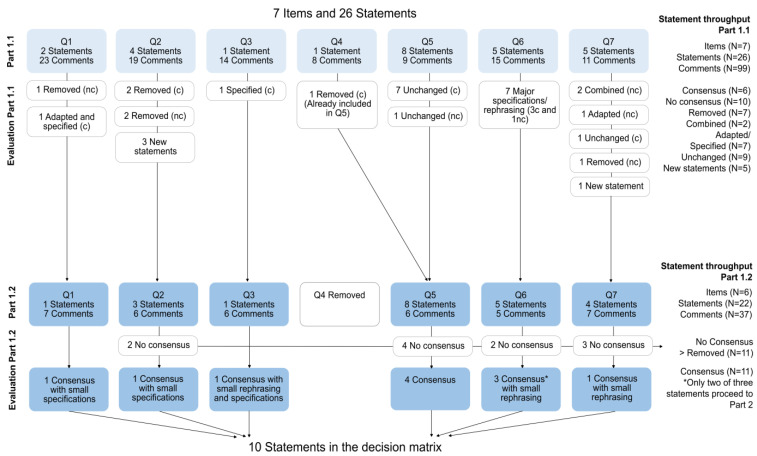
Item and statement throughput of the Delphi process in Part 1.1 and Part 1.2. Q = question/item, nc = no consensus, c = consensus * An exception was made for Q6 and Q7 (Appendix A) in which only the statements that indicate “the minimum number of publications” or “minimum number of patients” and reached consensus were added to the final list of statements.

**Table 1 IJNS-09-00056-t001:** Consensus and decision rules after each round.

Part 1.1/Round 1	Part 1.2/Round 2	Part 2/Round 3
Consensus was declared if a statement scored a mean of ≥7.0, a median of ≥7.0, and a mode of ≥7.0. Regardless of whether consensus was reached, statements proceeded to Part 1.2 to be discussed in an altered form.	Consensus was declared if a statement scored a mean of ≥7.0, a median of ≥7.0, and a mode of ≥7.0. Only statements reaching consensus proceeded to Part 2 ^1^.	A ≥ 75% majority agreement by the EP had to be reached to include a disorder in the NBS based only on *treatability*.
The RT was free to add small grammatical or textual changes to clarify the statement (also based on individual comments from the EP), provided that it did not compromise the meaning or implication of the statement.	The RT was free to add small grammatical or textual changes to clarify the statement (also based on individual comments from the EP), provided that it did not compromise the meaning or implication of the statement.	The RT was free to add small grammatical or textual changes to clarify the statement (also based on individual comments from the EP), provided that it did not compromise the meaning or implication of the statement.
If 3 or more EPs requested the same alteration, the statement was adapted accordingly.	No new statements were added after Part 1.2.	
If 3 or more EPs found the statement irrelevant, it could be removed, adapted, specified, or combined with another statement.		
The RT was free to add statements, based on the suggestions of individual panelists.		

^1^ An exception was made for Q6 and Q7 (Appendix A), in which only the statements that reached consensus that indicated the minimum number of publications or number of patients were added to the final list of statements. EP = expert panel, EPs = expert panel members, IMDs = inherited metabolic disorders, NBS = newborn screening, RT = research team.

**Table 2 IJNS-09-00056-t002:** Final statements defining *treatability* for inherited metabolic disorders.

Statement (S)/Question (Q)	
Q1./S1	There is a treatment available that is fully financially covered or reimbursed by standard health care (also when a patient reaches adulthood).
Q2./S2	The expected benefit/burden ratio of early treatment is positive and results in significant health benefits.
Q3./S3	The treatment/diet results in enough significant health benefits (compared to no treatment) to accept a small risk for disease-related mortality.
Q5.1./S4	Early detection with subsequent treatment, when compared to clinical presentation with subsequent treatment, prevents (sudden) death.
Q5.4./S5	Early detection with subsequent treatment, when compared to clinical presentation with subsequent treatment, prevents symptoms clearly related to one or two primary organ(s).
Q5.5./S6	Early detection with subsequent treatment, when compared to clinical presentation with subsequent treatment, prevents developmental delay.
Q5.7./S7	Early detection with subsequent treatment, when compared to clinical presentation with subsequent treatment, improves quality of life.
Q6.4./S8	Papers on treatments were published by at least two institutes with good quality data about a plausible mechanism with an “adequate number of patients” with clear effect size and suggesting the efficacy of early treatment is sufficient to accept that treatment for newborn screening.
Q6.5./S9	There is consensus on positive outcome of early detection by an (international) expert meeting.
Q7.3./S10	The expected effect of treatment was demonstrated in more than 75% of patients (both mild and severe variants).

**Table 3 IJNS-09-00056-t003:** Ten selected inborn metabolic disorders and assessment of *treatability* in different studies.

				Assessment of Genes in Studies (Yes/No) *				
	Name of Disorder	Associated Gene	OMIM ^2^	NGSf4NBS Project [18]	RUSP [7]	NEXUS [17]	Babyseq [16] Project	*Treatability* Delphi Study
IMD 1	PDE	*ALDH7A1*	266100	Yes	No	No	No	Yes
IMD 2	CG	*GALT*	606999	Yes	Yes	Yes	Yes	Yes
IMD 3	CPT2	*CPT2*	255110	Yes	Yes	Yes	No	No
			600649					
			608836					
IMD 4	GSD2	*GAA*	232300	No	Yes	Yes	Yes	No
IMD 5	GCH1	*GCH1*	233910	Yes	Yes	Yes	No	No
IMD 6	OTC	*OTC*	311250	Yes	No	Yes	Yes	No
IMD 7	WD	*ATP7B*	277900	Yes	No	No	Yes	No
IMD 8	MCM	*MMUT*	251000	Yes	Yes	No	No	Yes
IMD 9	TH	*TH*	605407	Yes	No	No	No	Yes
IMD 10	PKU	*PAH*	261600	Yes	Yes	Yes	Yes	Yes

* Yes/No: These genes were/were not included in similar studies, listed in this table, in which eligibility and genetic screening for newborn screening was tested. ^2^ OMIM = Online Mendelian Inheritance in Man. Every inborn metabolic disorder is identifiable by an OMIM code that can be used in databases. IMD = inherited metabolic disorder, NGSf4NBS = Next-generation Strategies First for Newborn Screening, RUSP = Recommended Uniform Screening Panel, NEXUS = North Carolina Newborn Exome Sequencing for Universal Screening study, Babyseq = The Babyseq Project. PDE = pyridoxine-dependent epilepsy, CG = classic galactosemia, CPT2 = carnitine palmitoyltransferase 2 deficiency, GSD 2 = glycogen storage disease type 2, GCH1 = autosomal recessive guanosine triphosphate cyclohydrolase 1 deficiency, OTC = ornithine transcarbamylase deficiency, WD = Wilson’s Disease, MCM = methylmalonic aciduria due to methyl malonyl-CoA mutase deficiency, TH = tyrosine hydroxylase deficiency, PKU = phenylketonuria.

**Table 4 IJNS-09-00056-t004:** Mean ranking scores ^1^ of the *treatability* statements per inherited metabolic disorder.

		PKU	CG	MCM	PDE	TH	CPT2	WD	GCH1	GSD2	OTC
Consensus on eligibility for NBS by the EP		Yes	Yes	Yes	Yes	Yes	No	No	No	No	No
		(−100.00%)	(−94.70%)	(−87.60%)	(−86.40%)	(−80.00%)	(−72.20%)	(−71.20%)	(−66.70%)	(−58.80%)	(−50.00%)
Response rate		N = 16	N = 19	N = 16	N = 22	N = 15	N = 18	N = 14	N = 12	N = 17	N = 16
S1		4.75	4.78	4.5	3.9	4.56	4.38	4.27	4.23	4.24	4.44
S2		4.88	4.42	3.94	4.05	4.19	3.88	3.93	4.08	4.06	3.63
S3		4.43	4.26	3.63	4.14	3.67	3.39	4.07	3.55	4.12	4
S4		2.79	4.22	3.69	3.4	2.87	3.56	2.92	2.33	3.59	3.94
S5		4.75	4.35	3.56	3.86	3.69	3.78	4.07	3.77	4.24	3.88
S6		4.75	3	3.56	3.43	3.69	3	3.83	3.38	2.88	3.44
S7		4.56	3.89	3.6	4.1	4	3.44	4.21	3.85	4.12	3.69
S8		4.88	4.32	3.75	4	3.21	3.22	3.54	2.67	4.12	3.75
S9		4.88	4.21	3.53	3.76	2.87	3	3.46	2.92	4.07	3.56
S10		4.75	3.95	3.13	3.9	3.21	3	3.21	3.17	3.56	3.38
Mean ^2^ per IMD		4.54	4.14	3.69	3.85	3.6	3.46	3.75	3.39	3.9	3.77

^1^ All scores are based on a 1–5 score on a Likert scale. ^2^ The mean per IMD is calculated from the means of all statements. IMD = inherited metabolic disorder, NBS= newborn screening, EP = expert panel, S = statement, PDE = pyridoxine-dependent epilepsy, CG = classic galactosemia, CPT2 = carnitine palmitoyltransferase 2 deficiency, GSD 2 = glycogen storage disease type 2. GCH1 = autosomal recessive guanosine triphosphate cyclohydrolase 1 deficiency, OTC = ornithine transcarbamylase deficiency, WD = Wilson’s Disease, MCM = methylmalonic aciduria due to methyl malonyl-CoA mutase deficiency, TH = tyrosine hydroxylase deficiency, PKU = phenylketonuria.

## Data Availability

Not applicable.

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
