# Peer review of "A Delphi Survey Study to Formulate Statements on the Treatability of Inherited Metabolic Disorders to Decide on Eligibility for Newborn Screening"

_2409-515X, 2023, doi:10.3390/ijns9040056_

Round 1

Reviewer 1 Report

IJNS Delphi W-J “Treatability”

Thank you for undergoing this important study and for sharing the results to benefit the international newborn screening community. The authors are to be commended for tackling these complex criteria that are more intertwining than many envision. This effort contributes a starting point on an iterative process to hone nuanced guidance. Most comments below are minor, none major, and some that this reviewer provides for greater consideration to be addressed in the manuscript and beyond. I am happy to review revisions, or clarify any details, should the authors or editors wish. 

Line 106 : “that elaborates on the W&J criterion treatability using a Delphi approach” – from this reader’s perspective, it would be clearer to italicize (alternatively, surround in quotes) the criterion term “treatability” and to do this consistently 

Line 149  “supported by 2 medical ethicists,” : please provide their initials here to parallel RT’s initials here (currently ethicists’ initials appear much later, and can keep there as well)

Lines 151-154 : to this reviewer, this sentence is challenging to follow. Consider placing “invited to this study…” earlier in the sentence

Line 217-218 : to this reviewer, this sentence is challenging to follow. : “….in which only the statements that reached consensus indicating the minimum amount of publications/ number of patients,…” Consider rearranging the wording and/or add a word between “consensus” and “indicating”

Line 246: to this reviewer, more specific content is needed here to provide context rather than having to move to an appendix. “Part 1.1 started with seven main items divided into a total of 26 statements (Appendix I).” Recommendation is for a brief table here that at least includes the Questions leaving out options wherever feasible. Appendix. 1 can have it all as does currently. 

Line 247-49: suggest breaking into two sentences to enhance readability as the points in this multi-compounded sentence get lost.

Line 248 : “suggestions on the ‘at least’ : boxes ... “ : within this sentence, please provide clarification of “at least”

Line 261 : please see similar comment Line 217-218

Table 2 : suggest inserting “final” in Table 2 header to align with Line 269 stating: “final list of statements is depicted in Table 2.”

Table 3: suggest inserting (by numerical superscript) the corresponding citation number from the reference list into Table 3

Line 371-372 : “Opinion 3: Concerns about how to proceed on disorders not suitable for NBS and/or disorders with broader phenotypic variability.” From this reviewer’s perspective, it would be beneficial to clarify/operationalize the term “suitability.” 

Consider also teasing these apart, at least in the discussion if not both here and the discussion

***Regarding Delphi method:

Related to the lines noted below: While this reviewer is an avid endorser and utilizer of the Modified Delphi approach for many of the reasons cited here, there are some important points worth considering here. There are many steps between design and reporting that can impact the validity and quality and that can introduce bias. As such ,  modification to lines 396-400+ are suggested. 

Line 396-97. “The validity and quality of a Delphi study however mainly depends on the design and reporting of the  Delphi by the investigator, rather than the technique itself

Lines 400-   “A risk of cognitive biases may occur…….

Additionally, the point below is an important consideration that this reviewer suggests needs to be addressed earlier in this paper as well.

Lines 419-20  “At the same time validation of the criteria in other groups of professionals e.g. international colleagues with experience in IMDs and patient representative organizations is needed.”

Importantly, to minimize study bias, ethicists can be beneficial as an integral member of the study team from the early design phase until all results are reported.

*** Regarding W&J criteria:

Related to the lines noted below: Please clarify why the authors chose to focus on the W-J specifically rather than incorporating “treatability” as used by others with perhaps more nuance, such as Andermann’s (while Andermann was listed in the references, her concept was not tied in here) From this reviewer’s perspective, the text (intro and discussion) would benefit from not just attaching the term “treatability” with W-J? Some examples in text:

Line 382 : “…not influenced by other W&J criteria”

Lines 386-87 “…create a scoring system on the W&J criterion on treatability”

**** These are important points, however to this reviewer the sentences beg the question, why promote distinctly separate criteria when in reality there is so much inter-relateness?  Recommend this is considered in the intro as well, and suggest a sentence noting that the W-J was not proposed for newborns.  Plus, important to mention that innovation generates new ideas about what is treatable.

Lines 423- 28: “Our statements were primarily based on the W&J criterion 2 on treatability. In principle, no other W&J criteria were considered. On the one hand, this is important as each  of the criteria should be evaluated separately in a transparent process. On the other hand, our study shows that it is hard to evaluate this criterion on its own as treatability links to other criteria such as costs and the question about the selection of patients who need to be treated, e.g. the age to become symptomatic or the variation in severity of presentation of disease.”…….

…….

Lines 432-36: “……the value of a single statement in the criterion of treatability can be low, e.g. statement 6 in GSD2, whereas the statements as a whole give a more complete impression of the level of treatability. Our data at least suggest that the transparency of the process was appreciated but that addressing the criterion treatability on its own necessitates addressing the other criteria separately in similar processes as well.”

……..

Lines 445-46: “…….stressing the fact that a panel of acknowledged disease experts for each disease is necessary to assess their eligibility for NBS.” This is an important point to this reviewer, and worthy of expanding to a few additional sentences.

***To this reviewer, important to state this early as well as here or seems like an afterthought.

Lines 467-76: “Further studies with other (international) groups of professionals (e.g. pediatric endocrinologists, and experts of IMDs) using similar scoring systems are needed to investigate the validity of our 10 statements and scoring system to ultimately decide on the eligibility of disorders for NBS. Such studies are also needed with parents (to be) and patient representatives. This may seem in contrast to the high level of expertise EPs considered  necessary to evaluate the statements, but the studies of Armstrong et al. 2022 from the Babyseq project and the study of Blom et al. 2021 provided at least some evidence that parents do understand the concept of choices whether to screen for (un)treatable disorders [40-42]. “

*** perhaps should not be so tied to W-J?(what about andermann?)

Line 477: “We built a scoring system based on the statements. This Delphi study allowed us to  gain insight into one specific W&J criterion (treatability).”

****Lines 482-488: “It is a starting point in selecting disorders for NBS in an era in which NGS techniques increase the number of IMDs and other genetic diseases technically eligible for NBS. Since this is the first study ever investigating a more transparent process for eligibility for NBS, this approach needs to be fine-tuned. We envision a funnel procedure to evaluate the eligibility of a disease by passing each of the W&J criteria one by one, following the same procedure for every criterion for every disease. 488

Consider adding a sentence about new era of innovative interventions/treatment re: “treatability.

Recommend inserting “TO OUR KNOWLEDGE” whenever referencing “first study”

And again, please clarify why stick to W-J? 

Line 532-33: Acknowledgements: “….medical ethicists Prof. Dr. W. J. (Wybo) Dondorp and 532

Dr. E. L. M. (Els) Maeckelberghe for their insights in the first draft of the statements.”

While important to include medical ethicists at early stages, also important to have bioethicists later too for interpreting the data and for blending points together to make a statement. -- to reduce all types of bias

Appendix 1 Initial Statements

To this reviewer, there seems to be a discordance between adjectives used in this statement.

Line 547 Q2 B : “Modest impact or burden with invasive screening tests, daily lifestyle/diet modification, and medication with a substantial side-effect profile”

Please clarify the decision made to align “modest impact” with “substantial side effect.” 

Moreover, please explain why the authors decided that IV treatment is considered more impactful than a medication that may lead to a substantial side-effect profile

Line 549 Q2 C: “Significant impact or burden including intravenous treatment”

Lines 553 -54 Q3 : “still carries a risk for disease-related mortality” Please clarify why there would be no optional adjective to modify level of risk (“significant”/high, versus low or modest/moderate) as had been done in Q2.

Lines 556 -57 Q4: “still carries a risk for disease-related morbidity” Please clarify why there would be no optional adjective to modify level of risk (high, versus low/moderat) as had been done in Q2.

This reviewer thanks the authors for including Appendix 3 data as often not considered as important as it is.

Line 641: Please add a space after A7. 

Line 959: Andermann A. Revisting wilson and Jungner in the genomic age: a review of screening criteria over the past 40 years. Bulletin of the World Health Organization. 2008;86(4):317-9.

Please include all the authors (unless format style does not permit.)

Author Response

Yours sincerely,

Abigail Veldman

Reviewer 2 Report

Summary 

This is a well written and topical paper which aims to address which IMDs might be considered applicable for NBS when treatability alone is used as the criterion. An online Delphi study was used to undertake this evaluation which is a novel approach.

The paper describes the methodology used for the study in good detail which will assist readers who are not familiar with this concept. One limitation of the paper is that the expert panel were selected from one country and although this point is made, it may be worth expanding on other limitations.

Table 4 summarises the mean ranking scores for each IMD. The likert scale of 0-5 was used here – does the range of scores provide any insight into the consensus between the different IMDs and if so, could the authors provide some commentary on this to assist the readers  (i.e the mean score per IMD ranges from 3.39 to 4.54 which seems quite close).

Author Response

Yours sincerely,

Abigail Veldman
